# 🥤 SODA: Million-scale Dialogue Distillation with Social Commonsense Contextualization

**Hyunwoo Kim**♡♠  **Jack Hessel**♡  **Liwei Jiang**♡◇  **Peter West**◇  **Ximing Lu**◇
**Youngjae Yu**♡  **Pei Zhou**♡♣  **Ronan Le Bras**♡  **Malihe Alikhani**†
**Gunhee Kim**♠  **Maarten Sap**♡‡  **Yejin Choi**♡◇

♡ Allen Institute for Artificial Intelligence  ♠ Seoul National University  ◇ University of Washington
♣ University of Southern California  † University of Pittsburgh  ‡ Carnegie Mellon University

## Abstract

Data scarcity has been a long standing issue in the field of open-domain social dialogue. To quench this thirst, we present 🥤 SODA: the first publicly available, million-scale high-quality social dialogue dataset. By contextualizing social commonsense knowledge from a knowledge graph, we are able to distill an exceptionally broad spectrum of social interactions from a large language model. Human evaluation shows that conversations in SODA are more consistent, specific, and (surprisingly) *natural* than those in prior human-authored datasets.

Using SODA, we train 🧑‍🚀 COSMO: a generalizable conversation model that is significantly more natural and consistent on unseen datasets than best-performing conversation models (e.g., GODEL, BlenderBot-1, Koala, Vicuna). Experiments reveal COSMO is sometimes even preferred to the original human-written gold responses. Additionally, our results shed light on the distinction between knowledge-enriched conversations and natural social chitchats. We make our data, models, and code public.[1]

## 1 Introduction

Conversations that occur in everyday spoken situations are often not recorded as data. And when they are, such as in the case of text messages, research use is rightly restricted due to privacy and legal concerns. As a result, collecting high-quality, everyday social conversations on a large scale has long been recognized as a difficult task (Smith et al., 2020). Previous studies have relied on crowdsourcing focused on specific themes of dialogue (e.g., persona, empathy; Zhang et al., 2018; Rashkin et al., 2019). However, this approach is limited in scale due to its associated costs. As a result, the progress made in machine dialogues, including generation, evaluation, and understanding, has been severely hindered by the reliance on these small datasets (Kann et al., 2022; Mehri et al., 2022).

Figure 1: An illustration of our $CO_3$ framework (§2), SODA dataset (§3), and conversation model COSMO (§4) trained on SODA. Conversations are distilled from a large language model (LLM) by contextualizing social commonsense. The full example is in Table 1.

To alleviate this bottleneck, we introduce 🥤 SODA (**SO**cial **DiA**logues), a million-scale English dialogue dataset covering a wide variety of social interactions. As a result of being grounded on rich social commonsense and narratives, SODA goes beyond specific skill-focused dialogues and features more general conversations. Our dataset includes 1.5 million dialogues distilled from a large language model (in our case, GPT-3.5; Ouyang et al., 2022) resulting in more than 11 million utterances with 300 million tokens: SODA is the largest publicly available open-domain social conversation dataset. Human evaluation shows that SODA surpasses existing human-authored dialogue corpora across axes like consistency, specificity, and (surprisingly, even) naturalness (§3.2).

To make SODA, we propose 🌐 $CO_3$, a framework for **CO**ntextualizing **CO**mmonsense for distilling **CO**nversations from a large language model

---

[1] https://hyunw.kim/sodaverse

(LLM). Illustrated in Figure 1, $CO_3$ infuses commonsense knowledge into dialogues by transforming knowledge triples into narratives, and then into dialogues. Such an approach offers two significant advantages: (1) maximizing diversity and (2) minimizing nonsensical conversations. Although generating content using LLMs is relatively easy, determining how to cover diverse content poses a non-trivial challenge. We find that sampling from an LLM without contexts results in dull conversations (§3.3). Because commonsense knowledge graphs cover a wide range of everyday situations (West et al., 2022), conditioning on them results in a broad spectrum of conversations. Moreover, since LLMs are prone to hallucinations (Weidinger et al., 2021), the seed commonsense knowledge can help them stay on a sensible generation path.

With SODA, we train a **CO**nver**S**ation **MO**del, COSMO. Human evaluation results demonstrate that: (1) COSMO generalizes better to unseen conversations than existing best-performing dialogue models, winning by more than 40% on average in head-to-head comparisons versus Blender-Bot (Roller et al., 2021), Koala (Geng et al., 2023), and Vicuna (Chiang et al., 2023) (§5.1); (2) COSMO outperforms BlenderBot (with the same number of parameters) *on the dataset Blender-Bot was trained on,* despite never seeing the corpus (§5.2); and (3) COSMO responses are even preferred over human-authored, ground-truth responses in DailyDialog (Li et al., 2017), a dataset on which COSMO was not trained on (§5.1).

Finally, the distilled dialogues in SODA represent a significant resource contribution for open-domain dialogue research. Most of all, SODA enables the research community to train smaller dialogue agents with competitive capabilities. Also, SODA can help enhance the generalizability of other advancements in the dialogue field (e.g., understanding and evaluation), which have relied on existing small datasets. Lastly, SODA highlights a dimension where recent LLM-based conversational agents (e.g., Koala, Vicuna, and ChatGPT) struggle – i.e., the naturalness of the responses (§5.1 and §5.3). As these models are designed to provide knowledge-based responses, they may generate responses that are informative but lack the naturalness found in social chitchat. We plan to publicly release SODA, COSMO, and $CO_3$ under the permissive license CC-BY-4.0, aiming to address the data scarcity issue in open-domain dialogue.

## 2 🗨️ $CO_3$: A Contextualization Framework for Conversation Distillation using Commonsense

We propose $CO_3$, a framework for distilling **co**nversations from large language models (LLMs) by **co**ntextualizing (i.e., adding more context information) **co**mmonsense knowledge. Our goal is to obtain natural conversations covering a wide variety of social interactions. $CO_3$ consists of three steps: (1) Retrieving social commonsense from a symbolic commonsense knowledge graph (§2.2), (2) converting it into sentence form and generating a narrative from the sentence (§2.3), and (3) inferring the conversation participants from the narrative and derive a conversation grounded in the narrative (§2.4). We use GPT-3.5 (i.e., text-davinci-002[2]; Ouyang et al., 2022) to implement $CO_3$, though in practice, a different model could be used. We use $CO_3$ to create SODA: an example is in Table 1. More details can be found in Appendix A.

### 2.1 Inspiration Behind $CO_3$

*What is at the heart of conversation?* At its core, a conversation is a fundamental form of social interaction (Myllyniemi, 1986). These experiences are abstracted into narratives or scripts (Mar and Oatley, 2008; Rumelhart, 1975; Schank and Abelson, 1975). Eventually, social experiences form our knowledge for explaining everyday events and inferring the mental states of others (Heider, 1958). This inference is coined *attribution* in social psychology (Baumeister and Bushman, 2017), and has been studied in NLP as *social commonsense* (Rashkin et al., 2018; Sap et al., 2019). Inspired by cognitive science, we reverse the abstraction process, starting from social commonsense knowledge in symbolic forms, and unfold rich narratives and conversations that could have initially encapsulated those commonsense knowledge.

### 2.2 Commonsense Knowledge Graph

Concretely, we start with a commonsense knowledge graph, which captures various relations of everyday events and inferences on others' mental states in symbolic forms (Sap et al., 2019; Hwang et al., 2021). The knowledge graph is represented by symbolic triples describing two events, denoted as the head and tail, and the relation between those two events, e.g., Head: `PersonX moves a step`

[2] https://beta.openai.com/docs/model-index-for-researchers/models-referred-to-as-gpt-3-5

closer to the goal, Relation: xNeed, Tail: to take the first step. We use Atomic[10x] (West et al., 2022) as our knowledge graph: it includes diverse social (e.g., intention, desire, reaction) and event-centered (e.g., order of events) commonsense. Since we are interested in distilling social interactions, we only retrieve triples related to *social* (rather than, e.g., physical) commonsense.[3]

## 2.3 Commonsense Knowledge → Narrative

**Triple Form to Sentence Form**   Since commonsense knowledge graphs are represented in symbolic form (i.e., triples), we first convert them into simple sentences with templates for each relation. For example, the commonsense knowledge in Table 1 is converted to "*Madeleine took the first step. Madeleine moves a step closer to the goal.*" To make the sentences sound more natural, we replace the person variables (e.g., PersonX, PersonY) with Top-1K common names of US SSN applicants ranging from 1990 to 2021.[4]

**Sentence Form to Narrative**   Next, we prompt GPT-3.5 to instantiate the sentence-form into a two- or three-sentence short narrative, e.g., for the commonsense example above " *Madeleine took the first step towards her goal, and with her coach's encouraging words, she moves one step closer.* " Prior work has shown that LLMs can effectively generate stories with plausible details that go beyond the contents of the prompt (Radford et al., 2019).

## 2.4 Narrative → Conversation

**Inferring Conversation Participants**   Inferring the conversation participants from the narrative is straightforward in cases where triples contain two person variables (i.e., PersonX and PersonY). But for triples that include only one person (e.g., the example in Table 1), we query GPT-3.5 to predict the other interlocutor (e.g., *mom*, *coworker*).

**Generating Conversation grounded in Narrative**   With the narrative and speakers as input, we prompt GPT-3.5 to generate a full, multi-turn conversation between the speakers in the context of the narrative. We append the first speaker as an utterance prefix to the prompt. Indicating the speakers with prefixes helps GPT-3.5 generate fluent conversations that alternate between the two.

---

[3]We leave relations for physical and event-centered commonsense to potential future work.

[4]catalog.data.gov/dataset/baby-names-from-social-security-card-applications-national-data

---

**Symbolic Social Commonsense Knowledge**:
- Head: PersonX moves a step closer to the goal
- Relation: xNeed
- Tail: to take the first step

**Sentence-form Commonsense**: Madeleine took the first step. Madeleine moves a step closer to the goal.

---

**Narrative**:
Madeleine took the first step towards her goal, and with her coach's encouraging words, she moves one step closer.
**Speakers**: Madeleine, Coach

---

**Conversation**:
**Madeleine:** Hey coach, I wanted to talk to you about my performance today. I was really pushing myself and I think I did pretty well. But I'm still not quite where I want to be.
**Coach:** Well Madeleine, you're progressing nicely. You've come a long way since we first started working together. But if you want to reach your full potential, there's still some work to be done.
**Madeleine:** I know that. And I'm willing to put in the work. It's just that sometimes I feel like I'm not making as much progress as I should be. Maybe I'm not training hard enough? Or maybe my technique is off?
**Coach:** It could be a number of things, Madeleine. But don't worry, we'll figure it out together. Let's just keep working hard and see how things go.
**Madeleine:** Alright, coach. Thanks for the talk.
**Coach:** No problem. See you at practice tomorrow.

---

Table 1: A sample from 🧋 SODA. More examples can be found in Appendix B.

# 3  🧋 SODA: A Million-scale Social Dialogue Dataset

We obtain SODA (**SO**cial **DiA**logues), a large-scale high-quality conversation dataset covering a wide range of social interactions, by applying a series of post-processing (§3.1) to the conversations generated from our contextualization framework (§2). We compare SODA with existing human-curated dialogue corpora (§3.2) and analyze the effectiveness of contextualization (§3.3). Table 1 shows a sample from our dataset. More details are in Appendix B.

## 3.1 Post-processing the Conversations

**Basic Filtering**   Starting with an initial set of 2.2 million conversations sampled from GPT-3.5, we: (1) use lexical pattern matching to filter out conversations with erroneous patterns – e.g., repetition and omission of speaker prefixes (6.3%); (2) remove conversations that have less than four turns or more than twenty turns (5.7%); (3) remove conversations with more than two speakers (11.3%);[5] and (4) remove conversations where at least one of the speakers was identified as non-human (e.g., broomstick, imaginary friend, dog; 5.6%).

---

[5]Although our pipeline naturally generates multi-party conversations as well, we focus on dyadic dialogues in this work.

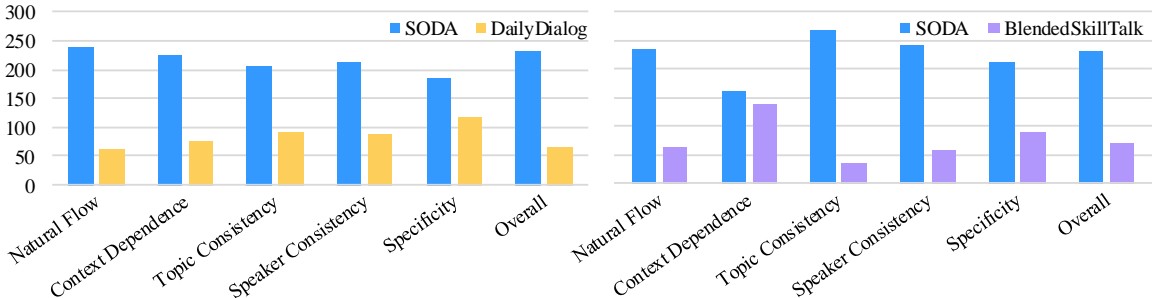

Figure 2: Results of head-to-head comparison between dialogues from 🥤 SODA, DailyDialog (Li et al., 2017), and BlendedSkillTalk (Smith et al., 2020) via human judgments (§3.2). The y-axis represents the number of samples preferred by human judges. The differences in all of the categories except for the *Context Dependence* comparing 🥤 SODA and BlendedSkillTalk are statistically significant ($|z| > 3.3$, $p < 0.05$).

**Safety Filtering** In order to avoid conversations with dangerous and harmful contents, we apply two safety filters: Canary (Kim et al., 2022a) and Rewire API.[6] Canary is a narrative dialogue safety model that can classify whether the given context needs caution or intervention. We discard all conversations marked as needing intervention (usually critical situations, e.g., crimes, emergencies; 4.3%); Rewire API is a web-based API for detecting toxic content. We discard all conversations that are above the threshold of 0.5 for any of the 'violence', 'hate', and 'sexually explicit' criteria (∼1%).

**Commonsense Filtering** We conduct a small-scale human evaluation via Amazon Mechanical Turk with 100 randomly sampled narrative-conversation pairs (3 annotators per instance) to check whether or not the seed commonsense triple is meaningfully instantiated by the narrative and conversation. According to majority vote, 88% of the instances include the seed commonsense knowledge. Given that the majority of human-annotated samples include the seed commonsense, we focus our filtering on excluding narrative-conversation pairs that lack the head event, as they are irrelevant to the given seed commonsense.

To apply this filter to all entries of the corpus, we use GPT-3.5 as a zero-shot classifier. As GPT-3.5 demonstrated great performance in question answering (Ouyang et al., 2022), we validate the generated narrative-conversation pairs by asking the language model itself to judge whether or not the head of the commonsense triple is implied. We formulate this as three-way multiple choice questions (i.e., *yes*, *no*, and *unknown*) and rank the answers according to their perplexity scores from GPT-3.5. This zero-shot classifier achieves high performance

on the human-annotated subset, with a precision of 97 for answering "yes". We find 95% of the filtered conversations are identified by GPT-3.5 as containing the head event. Pairs that lack the head event are removed to ensure relevance between the narrative-conversation pairs and commonsense triples. More details are in Appendix B.1.

**Final Dataset** After all filtering, 68.9% of the initial conversations remain, which form the 1,486,896 conversations in SODA.

**Name Bias Mitigation** We aim to minimize biases associated with specific names while increasing inclusion and diversity. Both language models and curated datasets often exhibit demographic imbalances (Dinan et al., 2020; Weidinger et al., 2021; Sheng et al., 2021). Inspired by Smith and Williams (2021), we randomly replace all names in conversations with Top-10K names of US SSN applicants from 1990 to 2021.[7] This covers 95% of all applicants' names from the chosen time range window, including various names from diverse gender[8] and ethnic backgrounds.

### 3.2 Comparing SODA with Human-authored Dialogues

**High Quality** To assess relative quality of the corpus, we conduct head-to-head human evaluations on Amazon Mechanical Turk, comparing SODA with two widely used open-domain dialogue datasets: DailyDialog (Li et al., 2017) and BlendedSkillTalk (Smith et al., 2020). We random sample 300 dialogues from each dataset and evaluate them according to six criteria (Mehri et al., 2022): (1)

---

[6] https://rewire.online/

[7] We use Top-1K names when contextualizing the commonsense triples in §2.3.

[8] Gender-neutral and nonbinary names are also included.

|  | #Dialog | Avg. #Turns | Avg. Utt. Length | Lexical Diversity |
|---|---|---|---|---|
| DailyDialog | 13K | 7.9 | 14.6 | 63.0 |
| PersonaChat | 11K | 14.8 | 14.2 | 43.6 |
| WizardOfWikipedia | 22K | 9.1 | 16.4 | 60.3 |
| EmpatheticDialogue | 25K | 4.3 | 13.7 | 64.2 |
| BlendedSkillTalk | 7K | 11.2 | 13.6 | 64.2 |
| ProsocialDialog | 58K | 5.7 | 20.0 | 60.2 |
| SODA | 1.5M | 7.6 | 16.1 | 68.0 |

Table 2: Statistics of SODA compared to other large-scale dialogue datasets. Utt. denotes utterance. Lexical diversity is measured with MTLD (McCarthy and Jarvis, 2010). Description for each dataset is in Appendix F.

natural flow, (2) context dependence, (3) topic consistency, (4) speaker consistency, (5) specificity, and (6) overall. Judges are asked to select a better dialogue between the two, regarding each criterion. For context dependence, we ask the judges to choose which conversation includes responses that are more dependent on previous turns. Further details are in Appendix B.2.

Despite being fully machine-generated, human raters judge SODA as better in quality compared to both DailyDialog and BlendedSkillTalk across all axes by a large margin, except for the context dependence comparing with BlendedSkillTalk (see Figure 2). In particular, evaluators rate the flow of SODA to be significantly more natural than other human-authored artificial conversation datasets.[9]

**Large Scale** With 1.5 million conversations, SODA is the largest in scale compared to existing crowdsourced open-domain dialogue datasets and the machine-human generated ProsocialDialog dataset (Table 2). It contains more than 11 million utterances and each conversation is grounded in a short narrative describing the context. In total, SODA consists of 300 million tokens, making it a rich source for training conversation models.

**Diverse Content** SODA is built on top of 1.5 million commonsense knowledge triples of Atomic[10x], which have been identified as being softly unique (West et al., 2022). Each seed triple is converted to a social narrative that serves as the distinct topic for each conversation. The Top-10 common keywords from these narratives are listed in Table 3.[10]

---

[9]A power analysis suggests that with our setup, we can detect effect sizes as small as 0.17 with a power and significance level of 95% (Faul et al., 2014).

[10]We prompt ChatGPT to output keywords of the narrative.

| Common keywords across all relations |
|---|
| friendship, help, support, communication, family, car, happiness, school, success, work |

| | Common keywords for each relation (excluding the above) |
|---|---|
| xAttr (18%) | kindness, anger, intelligent, responsibility, friend, trust, conversation, food, generosity, smart |
| xEffect (17%) | gratitude, anger, upset, hard work, happy, money, friend, boss, party, kindness |
| xIntent (23%) | independence, hard work, determination, money, relaxation, anger, kindness, store, understanding |
| xNeed (7%) | job, money, confidence, comfort, advice, interest, conversation, listening, store, park |
| xReact (25%) | frustration, anger, confidence, happy, pride, relief, disappointment, relaxation, anxiety, satisfaction |
| xWant (11%) | conversation, store, determination, apology, learning, doctor, job, friend, improvement, marriage |

Table 3: Common topic keywords of the narratives (i.e., conversation context) in SODA. Numbers in parentheses denote the ratio of the relations in SODA.

We find a broad spectrum of topics encountered in social interactions are included in SODA.

As a result, conversations in SODA contain diverse lexicons. We compute MTLD (McCarthy and Jarvis, 2010) to measure the lexical diversity of conversations. Table 2 reports the averaged diversity of dialogues for each training set. As PersonaChat (Zhang et al., 2018) contains conversations based on a few persona-related sentences, it shows the lowest lexical diversity. SODA, on the other hand, includes conversations from a variety of social situations, which leads to a wider range of words.

**Rich Emotion-related Information** Since commonsense knowledge from Atomic[10x] includes emotional reactions of people to events (i.e., the xReact triples), conversations with rich emotional contents are also included in SODA. In total, SODA includes 385K conversations generated from 1.7K unique emotion descriptions of the xReact triples' Tail (e.g., happy, ashamed, motivated, irritated).[11] Therefore, it contains significantly more descriptive emotion labels (i.e., the Tail) than other datasets which have fixed number of classes (Li et al., 2017; Rashkin et al., 2019). Furthermore, because we construct conversations in a bottom-up fashion from those emotion reaction in the commonsense triples, we know which speaker in the conversation is experiencing the emotion (i.e., PersonX) and what caused the emotion (i.e., the Head event).

---

[11]We note that conversations from other relations also naturally include emotional utterances.

| DailyDialog | | BlendedSkillTalk | | 🍵 SODA | |
|---|---|---|---|---|---|
| Emotion | Ratio | Emotion | Ratio | Emotion | Ratio |
| admiration | 20.42 | curiosity | 17.86 | curiosity | 12.92 |
| gratitude | 18.84 | admiration | 13.16 | admiration | 11.23 |
| curiosity | 12.85 | sadness | 8.50 | approval | 10.24 |
| approval | 10.91 | joy | 5.32 | gratitude | 7.39 |
| joy | 4.74 | excitement | 4.42 | joy | 6.38 |
| excitement | 3.61 | surprise | 4.34 | disappointed | 5.41 |
| surprise | 3.25 | disappointed | 4.34 | confusion | 4.68 |
| love | 3.06 | fear | 4.31 | surprise | 4.40 |
| optimism | 2.94 | approval | 4.19 | realization | 3.90 |
| caring | 2.23 | optimism | 3.95 | caring | 3.77 |

Table 4: The ratio (%) of Top-10 emotions in 10K utterances from DailyDialog, BlendedSkillTalk, and SODA, labeled by the GoEmotions' 27-emotion-type classifier (Demszky et al., 2020). Full table is in Appendix B.2.

We also find the distribution of emotions to be less skewed towards specific emotions. To compare the emotional composition, we use the 27-emotion-type classifier from GoEmotions (Demszky et al., 2020) for labeling and compare 10K utterances from DailyDialog, BlendedSkillTalk, and SODA. The distribution of emotions for each dataset is presented in Table 4. SODA exhibits a more balanced distribution of emotions while maintaining similar rankings with other human-authored dialogues.

**Cost & Time-Efficient**  Compared to dialogue crowdsourcing, collecting SODA via our contextualization framework is significantly more time and cost efficient. With GPT-3.5 text-davinci-002, to go from a commonsense triple to a dialogue costs about $0.02, and 10 queries take less than 2 minutes, counting our full filtration pipeline.

### 3.3 Do We Need Contextualization?

To isolate the effect of contextualization (vs. straightforward sampling from a large language model), we compare SODA with dialogues naively sampled from GPT-3.5 without any given context. We sample 100 dialogues using the same hyperparameters and the basic filtering steps in $CO_3$, but with the following prompt: "The following is a long in-depth conversation between two people.\nPerson 1:." We ask human judges to evaluate the conversations in a head-to-head comparison as before (§3.2), with the additional criterion of interestingness (See et al., 2019).

Figure 3 shows that judges significantly prefer context-grounded conversations. Conversations sampled without context are not only less specific and less interesting, but also exhibit lower lexical diversity than those from our $CO_3$ framework

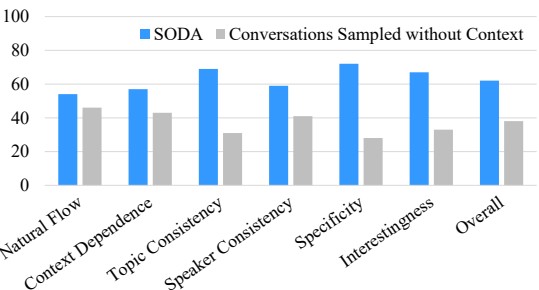

Figure 3: Results of head-to-head comparison human evaluation between conversations from SODA and those sampled from GPT-3.5 without context (§3.3). The y-axis indicates the number of samples that human judges preferred. The differences are all statistically significant with $|z| > 2.6$, $p < 0.05$ except for the *Natural Flow* class with $z = 1.1$ and $p > 0.05$.

(MTLD; McCarthy and Jarvis, 2010): 68.0 vs 63.1.

## 4 🛰️ COSMO: A Socially Situated Conversation Model

We use SODA to train **COSMO**: a **CO**nver**S**ation **MO**del that can converse in a wide range of social situations. COSMO can take in situation narrative, along with dialogue history, and generate a next utterance according to a given role.

**Training COSMO**  We use several structured components of SODA during training: (1) the contextual narrative $n$ (§2.3), (2) the perspective/speaker instruction $i$ (e.g., "*Imagine you are Madeleine and speak to her coach*") built with the inferred conversation participants (§2.4), and (3) the dialogue context $c$. The model is trained to generate a target response $r$ when given $n$, $i$, and $c$ – i.e., $p(r|n, i, c)$. We do so in a sequence-to-sequence fashion, concatenating $n$, $i$, $c$ with a separator <SEP> to serve as input. $c$ is made up of the previous conversation utterances concatenated with a turn indicator <TURN>.

Because conversational models often agree to toxic or unethical behavior (Baheti et al., 2021), for additional training data, we include Prosocial-Dialog (Kim et al., 2022a) (adapted to the same format as SODA, see Appendix C). ProsocialDialog includes a wide range of negative constructive feedback based on social rules-of-thumb, e.g., "*So I think it's best to continue being honest, and apologize that you were lying.*" The inclusion of this corpus assists conversation models in handling sensitive contexts (e.g., biased, harmful, unethical) without affecting the model performance on other datasets (Kim et al., 2022a).

We build COSMO on top of the LM-adapted T5 (Raffel et al., 2020; Lester et al., 2021), which achieves strong benchmark performance across various classification and generation tasks. (Sanh et al., 2021; Chung et al., 2022). We train two versions of the model: COSMO-3B and COSMO-11B using the T5X library (Roberts et al., 2022). For better robustness and generalizablity to datasets that don't have contexts or dialogue starting prompts, we randomly drop narrative $n$ and role instruction $i$ 30% and 50% of the time, respectively.

## 5 Generalizability of COSMO

We compare COSMO to other conversational agents on social conversation datasets under both out-of-domain and in-domain settings. Since automatic response evaluation is brittle, we focus on human evaluation (Smith et al., 2022). Automatic evaluation results via GPT-4 are in Appendix D.

**Baselines** We compare COSMO with four best-performing stand-alone conversation models: BlenderBot-1 (Roller et al., 2021), GODEL (Peng et al., 2022), Koala (Geng et al., 2023), and Vicuna (Chiang et al., 2023). BlenderBot is a transformer pretrained on 1.5B Reddit comments and trained on various chitchat datasets. GODEL utilizes a pretrained language model T5 (Raffel et al., 2020) trained on web text data, and further trains on 551M Reddit threads and 5M instruction and grounded dialogue datasets. Koala and Vicuna are models that finetuned LLaMA (Touvron et al., 2023), which is an open-source LLM, using dialogue data from the web. They are both known to achieve comparable performance to ChatGPT (OpenAI, 2022), which is a model finetuned for conversational interaction based on GPT-3.5 – i.e., our teacher model. We also compare COSMO with GPT-3.5 and ChatGPT; prompting details are in Appendix D.

**Evaluation Metrics** We perform head-to-head comparison between two responses, each from a different agent. We sample 100 test examples randomly from datasets and ask three human judges on Amazon Mechanical Turk to select the better response between the two in terms of four distinct criteria (Mehri et al., 2022): (1) naturalness, (2) consistency, (3) specificity, and (4) overall.

### 5.1 Out-of-domain Setting

We evaluate models on an unseen dialogue dataset, DailyDialog (Li et al., 2017), covering various daily

| Model | Natural | Consistent | Specific | Overall |
|---|---|---|---|---|
| BlenderBot-3B | 23% | 26% | 39% | 28% |
| COSMO-3B | **77%** | **74%** | **61%** | **72%** |
| GODEL$_L$ | 13% | 14% | 15% | 14% |
| COSMO-3B | **87%** | **86%** | **85%** | **86%** |
| Koala-7B | 30% | 34% | 30% | 29% |
| COSMO-3B | **70%** | **66%** | **70%** | **71%** |
| Vicuna-7B | 42% | 42% | 44% | 42% |
| COSMO-3B | **58%** | **58%** | **56%** | **58%** |
| Ground Truth | 43% | 45% | 46% | 45% |
| COSMO-3B | **57%** | **55%** | **54%** | **55%** |

Table 5: Results of head-to-head human evaluation between model responses on an unseen dataset: DailyDialog (Li et al., 2017) (§5.1). The differences are all statistically significant with $|z| > 12.45$ and $p < 0.05$, except for the *Specific* in the bottom row.

situations with emotions. Table 5 summarizes the head-to-head comparison results of the responses from COSMO and other models. Although COSMO is trained on significantly smaller amount of data (1.5M dialogues vs. 1.5B Reddit comments, 551M Reddit threads) and is significantly smaller (3B vs. 7B), it outperforms all other existing models with a significant margin across all aspects. Specifically, COSMO demonstrates the largest performance gap in terms of *naturalness*. It is worth noting that while Koala and Vicuna focus on providing informative responses, these results suggest that knowledge-seeking assistive conversations differ from natural social conversations.

In addition, we compare the responses from COSMO and 200 ground-truth responses in DailyDialog which were originally written by humans. Surprisingly, human judges prefer COSMO's responses even over the original gold responses in the dataset, suggesting that dialogue models trained on SODA can lead to high generalizability and naturalness, even for unseen conversations. Table 14 in the Appendix shows the ground-truth response and responses from each model for a given context.

### 5.2 One-sided Out-of-domain Setting

For an even harder setting, we evaluate COSMO vs. BlenderBot on the dataset BlenderBot was trained on: BlendedSkillTalk (BST; Smith et al., 2020). Table 6 (top) shows the head-to-head comparison results of the responses from COSMO and BlenderBot (for symmetry, we also evaluated BlenderBot on SODA with similar results; bottom row in Table 6). COSMO significantly outperforms BlenderBot on BST, its training domain (BlenderBot also

shows low performance on SODA). These results suggest that SODA contains patterns not present in existing datasets, but also covers patterns found in those datasets. More results are in Appendix D.

## 5.3 In-domain Setting

We also compare COSMO on SODA with its teacher GPT-3.5 and also ChatGPT, a chatbot-variant of the teacher.[12] Table 7 displays the head-to-head comparison results. In this setting, COSMO performs on-par with its teacher and ChatGPT, overall. In terms of specificity, COSMO's responses are significantly more specific than its teacher. Thus, SODA enables training competitive conversation models with a significantly smaller size (3B/11B) in comparison to existing large language models (175B).

Human judges evaluate ChatGPT's responses to be much more specific, but significantly less natural compared to COSMO. We hypothesize this is because ChatGPT is specially trained to give helpful and informative responses to user requests. Future work would be well-suited to compare the non-equivalence of simulating natural conversations vs. producing useful responses for users.

## 6 Related Work

**Building Dialogue Datasets with Large Language Models** Several studies have used large language models to augment or synthesize dialogue datasets. Zheng et al. (2023) and Chen et al. (2022) use GPT-J (Wang, 2021) to augment responses for emotional support conversations and understanding tasks, respectively. Chen and Yu (2021) trains a pseudo-labeler to increase the out-of-domain generalization of dialogue models. Ou et al. (2022) uses counterfactual reasoning to alter the semantics of responses and collect new ones. Kim et al. (2022a) proposes a human-machine collaborative framework, where a worker and GPT-3 take turns. Kim et al. (2022b) builds Blended Skill BotsTalk by letting multiple agents grounded in target skills engage for multi-skill dialogues. Chen et al. (2023) generate dyadic and multi-party conversations with topic words and show they have comparable quality to human-authored conversations. GPT-3 has also been used to help simulate task-oriented dialogues (Li et al., 2022) on a small scale. Others also augment dialogues with additional annotations – e.g., commonsense inferences (Zhou et al.,

---

| Model | Natural | Consistent | Specific | Overall |
|---|---|---|---|---|
| **BlendedSkillTalk** | | | | |
| BlenderBot-3B | 32% | 35% | 40% | 36% |
| COSMO-3B | **68%** | **65%** | **60%** | **64%** |
| **SODA** | | | | |
| BlenderBot-3B | 21% | 17% | 25% | 17% |
| COSMO-3B | **79%** | **83%** | **75%** | **83%** |

Table 6: Human evaluation results for head-to-head comparison of model responses under one-sided out-of-domain setting with COSMO and BlenderBot (Roller et al., 2021) (§5.2). BlendedSkillTalk (Smith et al., 2020) is an unseen dataset for COSMO, and SODA is an unseen dataset for BlenderBot. The differences are all statistically significant with $|z| > 4.24$ and $p < 0.05$.

| Model | Natural | Consistent | Specific | Overall |
|---|---|---|---|---|
| GPT-3.5 | **50%** | 46% | 31% | 47% |
| COSMO-11B | **50%** | **54%** | **69%** | **53%** |
| ChatGPT | 39% | 49% | **70%** | **50%** |
| COSMO-11B | **61%** | **51%** | 30% | **50%** |

Table 7: Head-to-head human evaluation between models on response generation for SODA (§5.3). The differences in the *Specific* from the top row, and the differences in the *Natural* and *Specific* from the bottom row are statistically significant with $|z| > 7.6$ and $p < 0.05$.

2022) or task-specific labels (Kulhánek et al., 2021; Chen et al., 2022). Compared to existing works, we are the first to contextualize commonsense knowledge graphs for generating narratives and derive full conversations from scratch in a significantly large-scale. This allows us to encompass an exceptionally broad spectrum of social interactions.

## 7 Conclusion

We presented 🥤 SODA, the first million-scale dialogue dataset covering an exceptionally wide range of social interactions to alleviate the data scarcity issue. SODA is not only orders of magnitude larger than popular dialogue datasets; it is also perceived to be significantly better than them across multiple aspects (e.g., naturalness, specificity, consistency). For making SODA, we also introduced 🧩 $CO_3$, a framework for distilling conversations from a large language model by contextualizing commonsense knowledge. With SODA, we trained a conversation model 🤖 COSMO that can generalize significantly better than existing models to unseen dialogues; and generate responses that are even more preferred than ground-truth responses of an existing dataset.

---

[12]Evaluation was run on the 2022 Dec 15 version: https://help.openai.com/en/articles/6825453-chatgpt-release-notes

# 8 Limitations

**Precautions taken during Dataset Construction**
Mining content from large language models might surface or even amplify harmful content within these models, such as biases and private information. With the goal of mitigating such danger, we take particular precautions to vet the safety of the distilled conversations.

First, previous studies have shown that human names commonly associated with certain gender and/or ethnicity result in biases in conversations produced by state-of-the-art dialog systems (Smith and Williams, 2021), such as BlenderBot (Roller et al., 2021). To diversify the name representations, we draw a wide range of common names representative of different gender and race identities from the US SSN name repository. Furthermore, to minimize potential harmful content from large language models, we filter generated dialogues by Canary, a dialogue safety detector model (Kim et al., 2022a), and Rewire API, a publicly available API for toxic content detection,[13] to remove dialogues with potentially toxic and dangerous content.

Our methods to pre-empt potential harmful content may not catch everything. For example, even with our diverse pool of names, there is still a focus on *common* names across gender and race, running the risk of misrepresenting marginalized groups. Similarly, no existing dialogue safety module or off-the-shelf toxicity detector is perfect at capturing all potentially harmful content. We strongly encourage future research along these directions to push the boundary of safe and responsible application usage of large language models.

During manual validation of commonsense and human evaluation, we compensate workers with an hourly wage of $15, which is over the US federal minimum hourly wage.

**Limitation of the Current Dataset and Future Work**  Here, we note some limitations of our work and suggest future directions. First, the dialogues in SODA are two-party only for now; because our framework also allows multi-party dialogue generation, we plan to explore this promising direction in the future.

Additionally, annotator biases might arise from the pool of annotators we recruit: we subselected annotators from a specific platform using specific filters which may cause unintended biases. We

hope future work will extend human evaluation to have potentially more annotator diversity.

Also, since SODA mainly focuses on social chitchat grounded on social commonsense, it lacks conversations grounded in scientific knowledge or historical facts. We seek to integrate other existing knowledge-grounded dialogue datasets into $CO_3$ in the future.

Finally, our choice of large language model (i.e., GPT-3.5) will likely affect the types of dialogues created. Future investigation may look into other potential large language model as sources to diversify the types and content of dialogues being generated. Similarly, future works can investigate other base models for COSMO that may lead to different quality of response generation.

**Intent of Technology and AI Regulation**  We want to stress that the intention of our work is *not* to build AI systems to replace humans. Instead, we want to build better assistive technologies, as chatbots are increasingly used in user-AI interactions and augmenting human-human conversations. Finally, to avoid situations where humans might be manipulated, we stress the need for improved regulations on the use and misuse of conversational AI systems (Crawford, 2021; Reich et al., 2021).

## Acknowledgement

We thank Jena D. Hwang for helpful discussions, and our colleagues on the Beaker Team at the Allen Institute for AI for helping with the compute infrastructure. This work was supported in part by DARPA MCS program through NIWC Pacific (N66001-19-2-4031). Hyunwoo Kim and Gunhee Kim are supported by the Institute of Information & communications Technology Planning & Evaluation (IITP) grant funded by the Korea government (MSIT) (No.2019-0-01082, SW StarLab; and No.2022-0-00156, Fundamental research on continual meta-learning for quality enhancement of casual videos and their 3D metaverse transformation). Lastly, we also thank OpenAI, as well as Google Cloud Compute.

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

# A Details of 🫧 CO₃

## A.1 Commonsense Knowledge → Narrative

**Retrieving Social Commonsense Knowledge**
We use the x-relations from Atomic[10x] (West et al., 2022), which are the inferences of people's mental states: xIntent, xWant, xReact, xAttr, and xNeed. Table 3 summarizes the ratio of relations included in our SODA dataset. We leave other relations (e.g., isBefore, isAfter) for future work.

**Triple Form to Sentence Form**    Table 8 lists the templates for converting symbolic commonsense knowledge to sentence form.

**Sentence Form to Narrative**    We prompt GPT-3.5 with "[sentence-form commonsense] Rewrite this story with more specific details in two or three sentences:". We find long narratives tend to be driven far away from the original commonsense knowledge. Therefore, we set the length of the narrative to two or three sentences.

We leverage text-davinci-002 GPT-3.5 for generating narratives. We set temperature to 0.9, top-p to 0.95, frequency penalty to 1.0, presence penalty to 0.6, and max tokens to 1024.

## A.2 Narrative → Conversation

**Inferring Conversation Participants**    We prompt GPT-3.5 with "[narrative] The following is a conversation in the scene between [PersonX's name] and ..." to let it finish the partial prompt. This yields a plausible interlocutor for a given narrative (e.g., *mom*, *classmate*, *coworker*, etc.); for the example story with Madeleine, "*her coach*" was predicted.

We leverage the text-davinci-002 GPT-3.5 model for identifying the speakers. We set temperature to 0, top-p to 1.0, frequency penalty to 0, presence penalty to 0, and max tokens to 16.

**Generating Conversation Grounded in Narrative**    We again leverage the text-davinci-002 GPT-3.5 model for generating conversations. An example prompt is "[narrative] The following is a long in-depth conversation happening in the scene between Madeleine and her coach with multiple turns.\nMadeleine:". We use the same hyperparameter setting as the narrative generation.

| Relation | Template for sentence form |
|---|---|
| xReact | [Head]. Now PersonX feels [Tail]. |
| xIntent | [Head] because PersonX wants [Tail]. |
| xAttr | PersonX is [Tail]. [Head]. |
| xEffect | [Head]. Now PersonX [Tail]. |
| xWant | [Head]. Now PersonX wants [Tail]. |
| xNeed | PersonX [Tail in past tense]. [Head]. |

Table 8: Templates for converting symbolic commonsense knowledge to sentence form.

| Relation | Template for building validation questions |
|---|---|
| xReact | Does PersonX feel [Tail] after [Head]? |
| xIntent | Does PersonX intend [Tail] when [Head]? |
| xAttr | Can PersonX be considered [Tail] when [Head]? |
| xEffect | [Head]. As a result, PersonX [Tail]. Is this true? |
| xWant | Does PersonX want [Tail] after [Head]? |
| xNeed | [Tail in past tense]. Is this true when [Head]? |

Table 9: Templates for converting symbolic commonsense knowledge to questions for validation.

# B Details of 🥤 SODA

Table 10 and Table 11 show samples from our dataset.

## B.1 Post-processing the Conversations

**Filtering Non-human Speakers**    First, we check whether the speaker prefix includes the name from our name base (§2.4). Next, we use lexical pattern matching and identify words in speaker prefixes that indicate humans (e.g., mom, dad, teacher, Mrs., Mr.). Finally, for speaker prefixes that do not match the above patterns, we prompt the text-davinci-002 GPT-3.5 model whether the speaker is human. For example, "Q: Is [speaker prefix] a person?\nA:"."

**Filtering with Commonsense Triples**    Using a prompt, we ask two questions about the Head event and also the Relation-Tail event for each instance: (1) is the head of the triple represented in the narrative-conversation pair; and (2) are the relation and tail? We prompt GPT-3.5 with "[narrative]\nQ: [head question]\nA:" and "[conversation]\nQ: [relation-tail question]\nA:" Table 9 lists the templates for

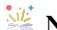 **Symbolic Social Commonsense Knowledge**:
- Head: PersonX provides another service
- Relation: xIntent
- Tail: to be a helpful person

**Sentence-form Commonsense**: Jabriel provides another service because Jabriel wants to be a helpful person.

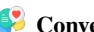 **Narrative**:
Jabriel provides a service by taking care of people's pets while they are out of town. Jabriel is a helpful person who loves animals, so she decided to start this business. She has been taking care of pets for three years and has never had any problems.

**Speakers**: Jabriel, Client

👫 **Conversation**:
**Jabriel**: Hi there! Are you looking for someone to take care of your pet while you're out of town?
**Client**: Yes, I am. I'm going on vacation for a week and I need someone to watch my cat.
**Jabriel**: That's no problem at all! I love taking care of animals, and I have experience with cats. I'll be happy to take care of your cat while you're gone.
**Client**: That sounds great. How much do you charge?
**Jabriel**: I charge $20 per day, plus an additional $10 per day if you want me to stay overnight.
**Client**: Sounds reasonable. I'll definitely need you to stay overnight, so it will be $30 per day. When can you start?
**Jabriel**: Any time that works for you! Just let me know when you'll be leaving and when you'll be back, and I'll take care of everything in between.
**Client**: Alright, I'll be leaving on Saturday and I'll be back next Saturday. Can you come over on Friday evening to meet my cat and get everything set up?
**Jabriel**: Sounds perfect. I'll see you on Friday at 6pm.

Table 10: A sample from 🥤 SODA.

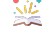 **Symbolic Social Commonsense Knowledge**:
- Head: PersonX takes on a lot of work
- Relation: xReact
- Tail: pressured

**Sentence-form Commonsense**: Yamir takes on a lot of work. Now Yamir feels pressured.

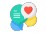 **Narrative**:
Yamir is a high school student who often takes on too much work. She frequently signs up for more extracurricular activities than she can handle and regularly agrees to do favors for her friends, even when she's already overwhelmed with other commitments. As a result, Yamir is constantly stressed out and feels like she can never catch up.

**Speakers**: Yamir, Lily

👫 **Conversation**:
**Yamir**: I can't believe I agreed to do this. I'm already so behind on everything else.
**Lily**: What's wrong? You seem really stressed out.
**Yamir**: I am. I feel like I can never catch up. Every time I turn around, there's something else that needs to be done. And it's all because I keep taking on too much.
**Lily**: Why do you keep doing that?
**Yamir**: I don't know. I guess I just feel like if I don't, then I'm not really doing my part or contributing enough. But it's just too much sometimes and it makes me so stressed out.
**Lily**: Well, maybe you need to start saying no more often. Just because someone asks you to do something doesn't mean you have to say yes. You have a right to say no and set your own limits.
**Yamir**: Yeah, maybe you're right...I should start saying no more often. It's just so hard though when people are counting on me and asking for favors all the time. But maybe if I start setting some boundaries, it'll help reduce my stress levels in the long run. Thanks for talking with me about this, Lily - it really helped put things into perspective!"

Table 11: Another sample from 🥤 SODA.

building questions for commonsense validation. For example, the commonsense knowledge triple in Table 1 will accompany questions of "*Madeleine moves a step closer to the goal, is this true?*" and "*Madeleine took the first step. Is this true when Madeleine moves a step closer to the goal?*" We formulate this as a three-way multiple choice question and rank answers (i.e., *yes*, *no*, and *unknown*) according to the perplexity score using conditional pointwise mutual information (Holtzman et al., 2021). We ask the questions with and without the context (i.e., the narrative and conversation). Table 9 lists the templates for building questions for commonsense validation. We find 66%, 95%, and 68% of filtered conversations are identified by GPT-3.5 as containing the full commonsense triple, the head event, and the relation-tail event, respectively: in total, 1,003,595 conversations are identified as fully encapsulating the seed commonsense knowledge.

Table 13 summarizes the performance of GPT-3.5 on 100 human-annotated samples for commonsense validation. We ask three human judges with the same question-answer format given to the model for each triple-narrative-conversation pair.

## B.2 Comparing SODA with Human-authored Dialogues

Figure 4 shows the annotation page for workers evaluating the dialogue quality.

**IRB Information** Crowdworking studies of standard NLP corpora (involving no personal disclosures) are not required by our IRB to be reviewed by them. While the authors of this work are not lawyers and this is not legal advice, this opinion is based on United States federal regulation 45 CFR 46, under which this study qualifies as exempt. We do not release crowdworker IDs, so annotations cannot be back-traced to individual workers.

| DailyDialog | | BlendedSkillTalk | | 🥤 SODA | |
|---|---|---|---|---|---|
| Emotion | Ratio | Emotion | Ratio | Emotion | Ratio |
| admiration | 20.42 | curiosity | 17.86 | curiosity | 12.92 |
| gratitude | 18.84 | admiration | 13.16 | admiration | 11.23 |
| curiosity | 12.85 | sadness | 8.50 | approval | 10.24 |
| approval | 10.91 | joy | 5.32 | gratitude | 7.39 |
| joy | 4.74 | excitement | 4.42 | joy | 6.38 |
| excitement | 3.61 | surprise | 4.34 | disappointed | 5.41 |
| surprise | 3.25 | disappointed | 4.34 | confusion | 4.68 |
| love | 3.06 | fear | 4.31 | surprise | 4.40 |
| optimism | 2.94 | approval | 4.19 | realization | 3.90 |
| caring | 2.23 | optimism | 3.95 | caring | 3.77 |
| remorse | 2.07 | realization | 3.84 | sadness | 3.76 |
| disapproval | 1.95 | annoyance | 3.48 | excitement | 3.20 |
| fear | 1.82 | love | 2.97 | remorse | 2.81 |
| sadness | 1.77 | confusion | 2.54 | disapproval | 2.74 |
| disappointed | 1.47 | caring | 2.31 | annoyance | 2.35 |
| annoyance | 1.41 | disgust | 1.99 | desire | 2.31 |
| confusion | 1.23 | nervousness | 1.88 | optimism | 2.23 |
| realization | 1.12 | remorse | 1.76 | love | 1.88 |
| anger | 0.97 | anger | 1.68 | fear | 1.81 |
| amusement | 0.92 | embarrassed | 1.44 | anger | 1.75 |
| desire | 0.89 | disapproval | 1.41 | nervousness | 1.45 |
| disgust | 0.51 | amusement | 1.09 | relief | 0.99 |
| nervousness | 0.27 | desire | 1.09 | embarrassed | 0.82 |
| embarrassed | 0.22 | pride | 0.74 | disgust | 0.58 |
| pride | 0.21 | gratitude | 0.66 | pride | 0.47 |
| relief | 0.21 | relief | 0.58 | amusement | 0.41 |
| grief | 0.00 | grief | 0.00 | grief | 0.00 |

Table 12: The ratio (%) of emotions in 10K utterances from DailyDialog, BlendedSkillTalk, and SODA, labeled by the 27-emotion-type classifier from GoEmotions (Demszky et al., 2020).

**Analysis on Emotion Distribution**  To obtain emotional responses, we randomly sample 10K utterances with emotion labels from DailyDialog (Li et al., 2017), utterances in conversations with the EmpatheticDialogue (Rashkin et al., 2019) theme for BlendedSkillTalk (Smith et al., 2020), and utterances in conversations generated from xReact triples for SODA. We run the finetuned BERT-base classifier (Demszky et al., 2020) on each utterance. Table 12 shows the full distribution across 27 emotion types for each dataset.

**Statistics of Human Evaluation**  A total of 74 workers participated in comparing dialogues, yielding a Krippendorf's alpha of 0.25. This indicates fair agreements on the quality judgments.

## C  Details of 🧩 COSMO

**Training Details**  COSMO-3B/COSMO-11B are trained using v3-32/v3-128 TPU accelerators with batch size 256 (effective batch ≈ 780) for 110K/130K additional steps using Adafactor (Shazeer and Stern, 2018) with constant learning rate .001.

|  | Precision | Recall | F1-score |
|---|---|---|---|
| **Head** | | | |
| Yes | 98.9 | 94.8 | 96.8 |
| No | 00.0 | 00.0 | 00.0 |
| Unknown | 16.7 | 100.0 | 28.6 |
| Overall | 96.1 | 93.0 | 94.2 |
| **Head /w PMI** | | | |
| Yes | 96.9 | 96.9 | 96.9 |
| No | 00.0 | 00.0 | 00.0 |
| Unknown | 00.0 | 00.0 | 00.0 |
| Overall | 94.0 | 94.0 | 94.0 |
| **Relation-Tail** | | | |
| Yes | 89.2 | 76.7 | 82.5 |
| No | 21.4 | 42.9 | 28.6 |
| Unknown | 8.3 | 14.3 | 10.5 |
| Overall | 78.8 | 70.0 | 73.7 |
| **Relation-Tail /w PMI** | | | |
| Yes | 92.2 | 68.6 | 78.7 |
| No | 21.4 | 42.9 | 28.6 |
| Unknown | 16.7 | 85.7 | 27.9 |
| Overall | 80.4 | 65.0 | 69.6 |

Table 13: Evaluation results of commonsense validation for short question-answering with InstructGPT on 100 human-annotated samples.

**Converting ProsocialDialog to SODA format**  We randomly sample names from our name database (§2.3) to construct the situation descriptions and perspective instructions for ProsocialDialog. The situation descriptions are made from the RoTs in ProsocialDialog (e.g., "*Cosmo is trying to gently convince a friend it's wrong to think all men are violent.*"); the instructions are built as we did for SODA (§4).

## D  Experiment Details

**Automatic Evaluation via GPT-4**  Inspired by Liu et al. (2023), we run automatic evaluation on the overall quality of responses with GPT-4. We use the same head-to-head comparison setup from Table 5 and 6 with the following prompt given to GPT-4: "You are a response evaluator. Your task is to choose the overall better response out of the two given the following context. You should consider naturalness, specificity, naturalness, and consistency.\n\nContext:\n{CONTEXT}\n\n1) {RESPONSE}\n2)    {RESPONSE}\n\nQuestion: Which response is better in terms of overall quality?\nAnswer: Response ".

**Dialogue Context**:

**A**: I really need to start eating healthier.

**B**: I have to start eating better too.

**A**: What kind of food do you usually eat?

**B**: I try my best to eat only fruits, vegetables, and chicken.

**A**: Is that really all that you eat?

**B**: That's basically it.

**A**: How do you stick to only those foods?

---

**Ground-truth response**:

Actually, fruits and veggies are really good for you.

---

**DialoGPT**: I eat a lot of fruit and veggies. I stick to a lot of things. I don't eat a lot of junk food.

---

**BlenderBot**: I eat a lot of fruit and veggies. I try to stay away from processed foods.

---

**GODEL**: I go to the gym and eat healthy food.

---

🌎 **COSMO**: I just try to make sure that I'm getting enough variety in my diet so that I don't get sick of eating the same thing all the time.

---

Table 14: The original ground-truth response and sample responses from DialoGPT, BlenderBot, GODEL, and COSMO to a context in DailyDialog.

| Dataset & Models | Overall |
|---|---|
| **DailyDialog** | |
| COSMO vs GODEL | **93%** vs 7% |
| COSMO vs BlenderBot | **68%** vs 32% |
| COSMO vs Koala | **65%** vs 35% |
| COSMO vs Vicuna | **54%** vs 46% |
| COSMO vs Ground Truth | **52%** vs 48% |
| **BlendedSkillTalk** | |
| COSMO vs BlenderBot | **66%** vs 34% |
| **SODA** | |
| COSMO vs BlenderBot | **85%** vs 15% |

Table 15: Automatic evaluation results of head-to-head comparison on overall quality of models' responses via GPT-4.

Table 15 shows the head-to-head comparison results for response quality between models. We find the results align closely with those from our human evaluation in §5. It should be noted that GPT-4 tends to favor GPT-generated texts over those written by humans, even when human judges show a preference for the latter (Liu et al., 2023). As a result, these scores are likely to be biased towards COSMO, which is trained on texts generated by GPT-3.5. Therefore, the original human evaluation

| Model | Natural | Consistent | Specific | Overall |
|---|---|---|---|---|
| **BlendedSkillTalk** | | | | |
| Koala-7B | 26% | 27% | 35% | 25% |
| COSMO-3B | **74%** | **73%** | **65%** | **75%** |
| Vicuna-7B | 43% | 47% | 45% | 46% |
| COSMO-3B | **57%** | **53%** | **55%** | **54%** |

Table 16: Human evaluation results for head-to-head comparison of model responses under zero-shot setting with COSMO, Koala (Geng et al., 2023), and Vicuna (Chiang et al., 2023). BlendedSkillTalk (Smith et al., 2020) is an unseen dataset for all three models.

results in Table 5 and 6 should be considered more significant when assessing the overall quality of the model, where COSMO also outperforms other models.

**Additional Human Evaluation on Blended-SkillTalk**   We also compare the response quality of COSMO, Koala (Geng et al., 2023), and Vicuna (Chiang et al., 2023) on BlendedSkillTalk (BST; Smith et al., 2020), which is an unseen dataset for all three models. We ask human judges to vote on which of the two model responses are better in terms of quality, based on four criteria as described in §5.2. Table 16 shows that COSMO outperforms both models in all four criteria, while the difference between COSMO and Vicuna is smaller compared to the difference between COSMO and Koala. Results on DailyDialog can be found in Table 5.

**Prompts for GPT-3.5, ChatGPT, Koala, and Vicuna**   We prompt GPT-3.5 with the following prompt: "You will be generating the next turn of a given dialogue between two people. Your response should be natural and specific. The dialogue is provided line-by-line.\n\ncontext:[narrative] \ndialogue:\n[dialogue]." For ChatGPT, Koala, and Vicuna, we use the following prompt: "You will be generating the next turn of a given dialogue between two people. Your response should usually be 1-2 sentences. Alongside the dialogue (which is provided line-by-line, where a new-line means the speaker changed), you'll be given some context about the two participants of the dialogue, e.g., their relationship, situation, etc.\n\n context:\n[narrative]\ndialogue:\n [dialogue]\nWhat is the most appropriate

next utterance (3 sentences max)?."

**Details of Human Evaluation**  A total of 77 workers participated in comparing responses, resulting in a Krippendorf's alpha of 0.5. This indicates good agreements on the response quality judgments. Figure 5 shows the annotation page for workers evaluating the response quality.

## E  Additional Related Work

**Human-authored Dialogue Datasets**  Existing dialogue datasets generally derive from one of the four sources: (1) Online learning websites and textbooks (Li et al., 2017) for beginners which may lack complex language usage. (2) Movie and drama scripts (Danescu-Niculescu-Mizil and Lee, 2011) that are less natural compared to day-to-day scenarios. (3) Crowdsourcing (Rashkin et al., 2019; Zhou et al., 2021; Tran et al., 2022): potentially prone to collecting responses that are somewhat short or dull due to incentive misalignment between researchers and crowdworkers (Zhou et al., 2022). (4) Noisy web interaction, such as Reddit comments (Baumgartner et al., 2020) and Twitter (Ritter et al., 2011); while widely used in dialogue agent pretraining stage due to their scale, these may represent different conversational frames compared to dyadic conversations. Moreover, as these are unfiltered conversations, their use surfaces a complex set of ethics and bias considerations. SODA contributes meaningfully to the suite of existing corpora via improved scale, quality, contextualization, and diverse commonsense knowledge.

## F  Dialogue Dataset Descriptions

DailyDialog is a dataset of casual dialogue compiled from English language learning websites (CC-BY-NC-SA-4.0; Li et al., 2017). PersonaChat is a dialogue dataset of two speakers getting to know one another based on provided personas (Zhang et al., 2018). EmpatheticDialogues contains empathetic conversations in which one speaker demonstrates empathy for the other speaker's emotions (Rashkin et al., 2019). Wizard of Wikipedia contains conversations based on Wikipedia between a speaker eager to learn and an expert speaker (Dinan et al., 2018). BlendedSkillTalk consists of conversations employing a variety of abilities – e.g., persona, empathy, knowledge (Smith et al., 2020). ProsocialDialog contains conversations where a speaker guides the interlocutor to follow social norms in problematic contexts (Kim et al., 2022a). Above datasets except for DailyDialog are all under the CC-BY-4.0 license. We use DailyDialog and BlendedSkillTalk for comparing with our SODA dataset, and ProsocialDialog for training COSMO, which is all compatible with the license.

We are studying meaningful **evaluation metrics** for the **qualities** of dialogues.

Specifically, you'll be given **two** pieces of dialogs, and you'll be asked to **compare which dialog is better** in terms of specific aspects, **specify which aspect was most important** for judging, and **write down your rationales in free-text**.

---

*Guidelines:*
1. **[Q1~6] First, choose which dialog is better regarding the given aspect.** There are four choices: **Definitely A/B** and **Slightly A/B** .
   - Please trust your instincts and choose **Definitely** if you would feel more confident giving one dialog, versus the other one.
   - Try to focus on quality over quantity. **Contentful/high-quality** dialog doesn't need to be lengthy.
2. **[Q7] Second, choose which aspect influenced you the most when judging the overall quality.**
   - If some factor other than the ones in Question 1~5 had the biggest influence, please select "Other" and specify.
3. **[Q8] Third, please describe in detail your option for the questions.**
   - It would be helpful to describe both *reasons you like the better dialog* **and** *reasons why you did not like the other dialog*.
   - Please be specific and detailed in your rationale.

*Note:*
- Please do not work on these HITs if you work at the

| Dialog A | Dialog B |
|---|---|
| ${dialoga} | ${dialogb} |

**Question 1.** Which dialog has a more **natural flow**?
 ○ Definitely A  ○ Slightly A  ○ Slightly B  ○ Definitely B

**Question 2.** Which dialog has more **back and forth engagement**? (more attentiveness / active listening)
 ○ Definitely A  ○ Slightly A  ○ Slightly B  ○ Definitely B

**Question 3.** Which dialog is more **consistent** and stays **on topic**?
 ○ Definitely A  ○ Slightly A  ○ Slightly B  ○ Definitely B

**Question 4.** Which dialog has **speakers** that are **less self-contradictory**?
 ○ Definitely A  ○ Slightly A  ○ Slightly B  ○ Definitely B

**Question 5.** Which dialog is more **specific**?
 ○ Definitely A  ○ Slightly A  ○ Slightly B  ○ Definitely B

**Question 6.** Which dialog has **higher quality overall**?
 ○ Definitely A  ○ Slightly A  ○ Slightly B  ○ Definitely B

**Question 7.** Which aspect affected you the most when judging the overall quality?
 ○ Natural flow  ○ Engagement  ○ Topic Consistency  ○ Speaker Consistency  ○ Specificity  ○ Other:

**Question 8.** Please justify, in detail, your answer for Question 1~7. What aspects of the better dialog **did** you prefer? Were there aspects of the worse advice you **did not** prefer?

Optional feedback?   (expand/collapse)

Figure 4: The annotation page for evaluating dialogues on Amazon Mechanical Turk.

We are studying meaningful **evaluation metrics** for the **qualities** of responses.

Specifically, you'll be given a piece of dialog and **two** responses, and you'll be asked to **compare which response is better** in terms of specific aspects, **specify which aspect was most important** for judging, and **write down your rationales in free-text**.

*Guidelines:*
1. **[Q1~4] First, choose which response is better regarding the given aspect.** There are four choices: Definitely A/B and Slightly A/B .
   ○ Please trust your instincts and choose Definitely if you would feel more confident giving one response, versus the other one.
   ○ Try to focus on quality over quantity. **Contentful/high-quality** response doesn't need to be lengthy.
2. **[Q5] Second, choose which aspect influenced you the most when judging the overall quality.**
   ○ If some factor other than the ones in Question 1~4 had the biggest influence, please select "Other" and specify.
3. **[Q6] Third, please describe in detail your option for the questions.**
   ○ It would be helpful to describe both *reasons you like the better response* **and** *reasons why you did not like the other response*.
   ○ Please be specific and detailed in your rationale.

*Note:*
- Please do not work on these HITs if you work at the University of Washington.

---

*Dialog Context*
${context}

| *Response 1* | *Response 2* |
|---|---|
| ${dialoga} | ${dialogb} |

Question 1. Which response is more **natural**?
◉ Definitely A   ○ Slightly A   ○ Slightly B   ◉ Definitely B

Question 2. Which response is more **consistent**?
◉ Definitely A   ○ Slightly A   ○ Slightly B   ◉ Definitely B

Question 3. Which response is more **specific**?
◉ Definitely A   ○ Slightly A   ○ Slightly B   ◉ Definitely B

Question 4. Which response do you like more **overall**?
◉ Definitely A   ○ Slightly A   ○ Slightly B   ◉ Definitely B

Question 5. Which aspect affected you the most when judging the overall quality?
○ Naturalness   ○ Consistency   ○ Specificity   ○ Other:

Question 6. Please justify, in detail, your answer for Question 1~4. What aspects of the better response **did** you prefer? Were there aspects of the worse response you **did not** prefer?

Optional feedback?   (expand/collapse)

Figure 5: The annotation page for evaluating responses on Amazon Mechanical Turk.