# OpenReview forum: "SODA: Million-scale Dialogue Distillation with Social Commonsense Contextualization"
_EMNLP/2023/Conference — EMNLP 2023 Main_

### Official Review · Reviewer_FFJY · 2023-08-04

**Soundness:** 4

**Excitement:**

3: Ambivalent: It has merits (e.g., it reports state-of-the-art results, the idea is nice), but there are key weaknesses (e.g., it describes incremental work), and it can significantly benefit from another round of revision. However, I won't object to accepting it if my co-reviewers champion it.

**Paper Topic And Main Contributions:**

This paper proposes CO3, a framework for automatically generating high-quality conversations based on LLMs (text-davinci-002 in this paper). Specifically, CO3 first sample commonsense knowledge triples from a knowledge graph (Atomic) and then uses the sampled commonsense knowledge to guide conversation generation. Based on CO3, this paper constructs SODA, a million-scale commonsense grounded English dialogue dataset. Furthermore, the authors use SODA to train a dialogue model COSMO. Experiment results demonstrate the effectiveness of the automatically constructed dialogue data.

**Questions For The Authors:**

1. Is grounded information (e.g., persona and knowledge in BST) leveraged in evaluations when comparing COSMO with other models?

2. Why does COSMO-11B surpass its teacher model InstructGPT and ChatGPT in several dimensions? What if these LLMs are prompted in the CO3 framework to generate responses?

3. How are the dialogue contexts used for evaluation (in Section 5.1, 5.2, 5.3) sampled?

**Reasons To Accept:**

1. The scarcity of large-scale open-domain dialogue has been a long-standing problem. Recently, generating high-quality data by prompting LLMs has been widely adopted in various domains (e.g., knowledge graph, instruction-following data, tool-use data, emotional support dialogue data). This paper extends this framework to the task of open-domain dialogue.

2. This paper leverages commonsense knowledge triples to prompt LLMs. Experiment results in Section 3.3 verify the effectiveness of this design.

**Reasons To Reject:**

1. Lack of automatic evaluation results. I understand the difficulty of automatically evaluating open-domain dialogue models with previous metrics (e.g., BLEU, ROUGE, BERTScore).  I would recommend conducting evaluations using ChatGPT or GPT-4, which has been widely adopted in recent works, including open-domain NLG.

2. Lack of human evaluation details. While this paper only consists of human evaluations, it lacks lots of human evaluation details. For example, how many annotators are recruited to annotate per instance, and what is the inter-annotator agreement. Additionally, based on the annotation page shown in Figure 4 and Figure 5, some evaluation dimensions are not well-defined. For example, naturalness, which is claimed as the key difference between COSMO and other baseline models, is simply illustrated as "Which dialog has a more natural flow" or "Which response is more natural?". I would recommend further clarifying these definitions, not only to researchers but also to annotators.

3. The training setting for COSMO is confusing. ProsocialDialogue is augmented into the original SODA dataset, which is claimed to assist models in handling sensitive contexts. However, the safety of COSMO is not evaluated in experiments. I recommend conducting evaluations on the model solely trained on SODA.

4. The evaluation setting between SODA and the previous dataset is unreasonable. Considering they do not share the same prefix or the same topic, it is hard to tell which one is better. Therefore, I recommend independently conducting evaluations on each dataset instead of pair-wise comparisons.

**Reproducibility:**

3: Could reproduce the results with some difficulty. The settings of parameters are underspecified or subjectively determined; the training/evaluation data are not widely available.

**Reviewer Confidence:**

4: Quite sure. I tried to check the important points carefully. It's unlikely, though conceivable, that I missed something that should affect my ratings.

---

> ### Author Rebuttal · Authors · 2023-08-29
>
> We thank you for your appreciation of the effectiveness of leveraging commonsense knowledge graphs!
>
> **1. Automatic evaluation results using GPT-4:**
> Thank you for the suggestion! We primarily relied on human evaluation due to the known limitations of automatic evaluation, even when using GPT models [1]. However, in response to your suggestion, we ran automatic evaluation on the overall quality of responses with GPT-4, under the same evaluation setup in Table 5 and 6. We find the results align closely with those from our human evaluation.
>
> |      DailyDialog      |    Overall   |
> |-----------------------|:------------:|
> |     COSMO vs GODEL    |  **93%** vs 7%   |
> |  COSMO vs BlenderBot  |  **68%** vs 32%  |
> |    COSMO vs Koala     |  **65%** vs 35%  |
> |    COSMO vs Vicuna    |  **54%** vs 46%  |
> |    COSMO vs GT        |  **52%** vs 48%  |
>
>
> |    BlendedSkillTalk   |    Overall   |
> |-----------------------|:------------:|
> |  COSMO vs BlenderBot  |  **66%** vs 34%  |
>
> |         SODA          |    Overall   |
> |-----------------------|:------------:|
> |  COSMO vs BlenderBot  |  **85%** vs 15%  |
>
> It should be noted that GPT-4 tends to favor GPT-generated texts over those written by humans, even when human judges show a preference for the latter [1]. As a result, these scores are likely to be biased towards COSMO, which is trained on texts generated by GPT-3.5. Therefore, our original human evaluation results should be given much more weight when assessing the overall quality of the model, where COSMO also outperforms other models.
>
>
> **2. Statistics of the human evaluations and definition of ‘naturalness’:**
> Thank you for the good points! A total of 74 workers participated in comparing dialogues, yielding a Krippendorf’s alpha of 0.25. For response comparisons, 77 workers participated with a Krippendorf’s alpha of 0.5. These results indicate fair to good agreements on the quality judgments.
>
> We will clarify the definition of the ‘naturalness’ aspect, which was used to determine which sounds more like a natural ‘human’ conversation or response [2]. We analyzed the workers’ responses for the rationales behind their judgment and also found that many of them mentioned human-like naturalness in their responses. For example, “(option) A seems more natural and like the conversation is just a real one between two people.”, “B feels more real to me like a conversation between real people.”, “A is more natural, whereas B sounds mechanical”, and “B is a more human and natural reply.” For reference, the Krippendorf’s alpha of ‘naturalness’ for the comparison of dialogues and responses, was 0.23 and 0.49, respectively, also indicating fair to good agreements across workers.
>
>
> **3. Training setup for COSMO and safety:**
> We included the ProsocialDialog to improve COSMO’s robustness to unsafe dialogue contexts in anticipation of public release. We ran a zero-shot experiment with COSMO on ToxiChat following the original ProsocialDialog work. We find COSMO achieves comparable performance with their original model Prost and InstructGPT, in terms of disagreeing with toxic dialogue input and avoiding generating offensive responses or toxic words: disagree (10.3), agree (7.5), offense (5.8), and bad (5.1).
>
> Furthermore, considering that ProsocialDialog constitutes only 3.7% of COSMO’s training data and our current evaluations do not explicitly include unethical dialogues, its impact on our existing evaluations is likely negligible. The original ProsocialDialog work also reports that their dataset is orthogonal to other datasets.
>
> **4. Comparison of "head-to-head" vs. "independently-evaluated scores" when conversation topics vary:**
> We appreciate your suggestion! We also debated the same question when conducting the evaluations in our paper. Unfortunately, both evaluation methods have limitations when the conversation topics are different. We acknowledge that comparing independently-evaluated scores may seem to provide a more isolated assessment of each dialogues, but this method can also be noisy and unreliable. As there is no direct comparison, the evaluators can hold different standards or expectations for different topics, making it complicated to compare them directly [3, 4, 5]. Given that our primary object was to directly compare the dialogue quality, we opted for a head-to-head comparison. We would like to note that we also ask workers to provide rationales for their judgments to minimize hasty comparisons during evaluation.
>
>
> **5. Is grounded information leveraged in evaluations when comparing COSMO with other models?**
> Yes, we provide the persona information to BlenderBot when conducting comparison for BlendedSkillTalk (Table 6), but not the ground-truth external knowledge sentences. As COSMO is not trained using the BlendedSkillTalk-formatted information, we only provide the dialogue history to COSMO. Since DailyDialog does not provide grounding information, we only feed the dialogue history to the models for the experiments in Table 5.
>
>
> **6. Why does COSMO-11B surpass its teacher model InstructGPT and ChatGPT in several dimensions? What if these LLMs are prompted in the CO3 framework to generate responses?**
> This is because COSMO is specifically trained on specific natural dialogues, yet ChatGPT etc. are general-purpose LMs. While CO_3 is primarily a data generation framework rather than a prompting method for response generation, those LLMs should perform as well as (or even better than) COSMO when given the same prompt containing all the information from the CO_3 framework, such as the ground-truth narrative and speakers.
>
>
> **7. How are the dialogue contexts used for evaluation (in Section 5.1, 5.2, 5.3) sampled?**
> We randomly sample 100 contexts from the test set.
>
> [1] Liu, Yang, et al. "G-Eval: NLG Evaluation using GPT-4 with Better Human Alignment" arXiv preprint arXiv:2303.16634 (2023).
>
> [2] Matthew Marge, João Miranda, Alan Black, and Alexander Rudnicky. 2010. Towards Improving the Naturalness of Social Conversations with Dialogue Systems. In Proceedings of the SIGDIAL 2010 Conference, pages 91–94, Tokyo, Japan. Association for Computational Linguistics.
>
> [3] Liang, Weixin, James Zou, and Zhou Yu. "Beyond User Self-Reported Likert Scale Ratings: A Comparison Model for Automatic Dialog Evaluation." Proceedings of the 58th Annual Meeting of the Association for Computational Linguistics. 2020.
>
> [4] Anja Belz and Eric Kow. 2010. Comparing Rating Scales and Preference Judgements in Language Evaluation. In Proceedings of the 6th International Natural Language Generation Conference. Association for Computational Linguistics.
>
> [5] Sashank Santhanam and Samira Shaikh. 2019. Towards Best Experiment Design for Evaluating Dialogue System Output. In Proceedings of the 12th International Conference on Natural Language Generation, pages 88–94, Tokyo, Japan. Association for Computational Linguistics.

---

### Official Review · Reviewer_1GY1 · 2023-08-11

**Soundness:** 4

**Excitement:**

4: Strong: This paper deepens the understanding of some phenomenon or lowers the barriers to an existing research direction.

**Missing References:**

[1] Xiang J, Liu Z, Zhou Y, et al. ASDOT: Any-shot data-to-text generation with pretrained language models[J]. arXiv preprint arXiv:2210.04325, 2022.

[2] Li D, Li Y, Zhang J, et al. C3kg: A chinese commonsense conversation knowledge graph[J]. arXiv preprint arXiv:2204.02549, 2022.

**Paper Topic And Main Contributions:**

In this paper, they propose a data synthesis method to distill dialogue using a large language model and commonsense knowledge graph. They do so to overcome data scarcity issues in the open-domain dialogue field.

Specifically, they follow certain templates to convert triplets to sentences first. Then, they prompt GPT-3.5, a large language model, to generate a paragraph of narrative according to the sentence. With the generated narratives, they prompt GPT-3.5 again to produce multi-turn dialogue. After obtaining the synthesis corpus, they conduct a series of post-processing to guarantee the quality of the dataset and remove harmful content, including rule-based filtering, safety filtering and commonsense filtering. The final dataset, SODA, contains about 1.5 million dialogues. Several evaluations are done subsequently to evaluate the quality, scale, diversity and emotional richness of their proposed dataset. They also conduct an ablation study to validate the effectiveness of their contextualization process in the synthesis pipeline.

To utilize the collected dataset, they train a new conversation model called COSMO based on it. They adopt human evaluation to compare COSMO with other chatbot and find COSMO outperform its competitors in various testing settings with smaller model size.

**Questions For The Authors:**

1. Have you tried to sample a set of relevant triplets from ATOMIC to generate a round of dialogue?

**Reasons To Accept:**

1. The idea of combining large language models with commonsense knowledge graph to do data synthesis sounds novel and useful. While this paper validates its effectiveness in dialogue dataset synthesis, this concept could also be applied in other areas, like table-to-text generation, to build high-quality and large-scale datasets.
2. The data synthesis pipeline in this paper is well-built and thoughtful. They consider various aspects including quality, safety, naturalness and consistency. Evaluation and statistics of their collected dataset are also comprehensive.
3. COSMO trained with SODA demonstrates promising performance. It outperforms the previous SOTA models of dialogue generation in different evaluation scenarios, with an even smaller size. It also shows comparable performance to GPT3.5 and ChatGPT, which contain much more parameters than it.
4. They will open-source the dataset and the model, and it would be useful for the community.

**Reasons To Reject:**

1. The major issue I am worried about is the complexity of SODA. Here I use complexity to refer to the complexity of the core event in each dialogue. This is because SODA is built by contextualizing and rewriting exactly one triplet into a multi-turn dialogue. However, each dialogue in people’s everyday life could involve more than one topic and thus contains more complex dialogue flows. Also, some triplets in ATOMIC are very simple and vague, which may further reduce the topical variety and complexity of SODA.
2. The authors highlight the rich motion-related information contained in SODA in section 3.2. But I don’t see any emotion-related evaluation in their experiment section to evaluate COSMO. Actually, the three metrics they used (natural, consistent and specific) are a little general and thus may miss some aspects like emotion and empathy.

**Reproducibility:**

4: Could mostly reproduce the results, but there may be some variation because of sample variance or minor variations in their interpretation of the protocol or method.

**Reviewer Confidence:**

4: Quite sure. I tried to check the important points carefully. It's unlikely, though conceivable, that I missed something that should affect my ratings.

---

> ### Author Rebuttal · Authors · 2023-08-28
>
> Thank you for recognizing the novelty of our approach in combining commonsense knowledge graphs with LLMs and its potential applications across various domains in the field! We also appreciate the acknowledgment of COSMO's performance.
>
> **1. The topical diversity and complexity of SODA:**
> We fully agree that this is an important point! The reason why we use step-by-step contextualization in our CO_3 framework is to tackle this exact issue. We observed the lack of specificity and diversity when we naively distill conversations directly from LLMs, as shown in Figure 3. Thanks to the contextualization steps (triple → short narrative → conversation), we find a greater diversity of commonsense knowledge and improved specificity being incorporated with each step. Additionally, we use Atomic10x instead of the original Atomic, which is significantly larger than Atomic. As a result, we find conversations in SODA exhibit a much higher lexical diversity compared to existing datasets (Table 2).
>
> **2. Emotion related experiments with COSMO:**
> Thank you for the nice suggestion! Although we did not directly test the emotion-related capabilities of COSMO, we would like to emphasize that DailyDialog and BlendedSkillTalk datasets both include emotion-annotated contexts with 6 and 32 types of emotion, respectively. The fact that COSMO outperforms existing models and even the ground truth response suggests that it can well adapt to the context’s emotion. We will add this discussion in the updated draft.
>
> **3. Sampling a set of relevant triplets from ATOMIC to generate a round of dialogues:**
> Thank you for the thoughtful suggestion! In our early experiments, we found it was difficult to automatically sample multiple commonsense triples that lead to natural narratives and conversations. However, we agree that this is an interesting future direction.
>
> **4. Missing references.**
> Thank you, we will include these in the updated draft.

---

### Official Review · Reviewer_Czz8 · 2023-08-11

**Typos Grammar Style And Presentation Improvements:** N/A
**Soundness:** 4

**Excitement:**

4: Strong: This paper deepens the understanding of some phenomenon or lowers the barriers to an existing research direction.

**Paper Topic And Main Contributions:**

*SODA: Million-scale Dialogue Distillation with Social Commonsense Contextualization* consists of three major contributions: SODA, CO_3, and COSMO. SODA is a million scale social dialogue dataset rooted in the narrative of social common sense knowledge triplets, CO_3 is the pipeline for contextualizing the commonsense triplets, and COSMO is the resulting conversational model trained on the SODA dataset. The dataset outranks Daily Dialog and BlendedSkillTalk in natural flow, context dependence, topic and speaker consistency, and specificity per AMT workers. SODA manages to include a diverse range of content including emotion related information despite coming from common sense triplets. Even though SODA uses GPT 3.5 as its teacher, the authors demonstrate that the dialogs from SODA (using the CO_3 framework), outperforms the teacher network (without context) GPT 3.5. The authors train COSMO with is a LM-adapted T5 and results in better performance at a fraction of the price of GPT 3.5  (3B/11B parameters vs 175B parameters). The authors evaluate COSMO on an out of domain dataset (Daily Dialog) and comparable models Blender, GODEL, Koala, Vicuna, and Human gold values. COSMO significantly outperforms all models across naturalness, consistency, and specificity with the largest difference found in naturalness (87% over Blender 23%, GODEL 13%, Koala 30%, Vicuna 42%, and even Human Gold 43%). They pitted COSMO against BlenderBot on the BlendedSkillTalk dataset, a training set BlenderBot is trained on, and outperform BlenderBot significantly across the same metrics. Finally, the in domain setting is tested by evaluating COSMO, GPT 3.5 and ChatGPT on SODA; the SODA trained COSMO performs best.

**Questions For The Authors:**

- Q1: I did not see GPT 3.5 or ChatGPT evaluated in Table 5 Daily Dialog. The in-domain performance of GPT 3.5 on naturalness and consistency (Table 7) begs the question on Table 5, *"Do you think COSMO scores are elevated in Table 5 by the underlying conversations generated from GPT 3.5 in SODA dataset?"

- Q2: Can we expect GPT 3.5 to perform comparably to COSMO in naturalness and consistency in Table 5 (from Table 7 performance) thus having a similar improvements over Blender, GODEL, Koala, Vicuna, and GT?*

- Q3: I was pleased to see a listed limitation, namely, *"SODA mainly focuses on social chitchat grounded on social commonsense, it lacks conversations ground in scientific knowledge or historical fact"*. Grounding backstories is a core principle of the SODA dataset because of the CO_3 framework. Do you think the metrics selected (Natural, Consistent, Specific) favor SODA's characteristic of detailed grounding backstories? However, humans must frequently inquire about missing backstory information (not necessarily scientific or historical information). Thus, is SODA representative of human-bot or human-human conversations? This can affect the applicability of SODA.

- Q4: By my calculations @ 1.5M dialogues keeping 68.9% *viz.* 2.2M dialogues @  \\$.02 per dialogue - without overhead -  costs puts SODA at approximately \\$43,500. Is this an accurate representation of the cost for performance/cost comparisons?

- Q5: Why is the discrepancy between humans and GPT 3.5 so large in identifying if the dialogue contains the head event (88% vs 95%)? Do you think GPT 3.5 is over estimating its performance? How large is the holdout for this evaluation? Only 100 dialogues?

**Reasons To Accept:**

- Large and costly effort for dialogues distilled from narratives and social commonsense.

- Diverse dialogues, potentially helpful in a large number of chatbot applications.

- Convincingly sound evaluations with AMT workers, the improvements for out of domain evaluation Table 5 was significant and noteworthy.

- COSMO outperforms GPT 3.5 and ChatGPT and will likely serve as a baseline in future evaluations.

- SODA dataset appears to be better in reported metrics by significant margin to DailyDialog and BlenderSkillTalk; similar open domain social dialogue datasets.

**Reasons To Reject:**

- AMT workers are notoriously inferior to in-domain participants, e.g. computational linguist graduate students, thus ill-suited for any rigorous evaluation. An evaluation with domain aware participants and a criteria that does not favor a narrative based approach might yield closer results for Table 5. To be clear I do not think the participant or metric selection neutralizes the demonstrated improvements in any significant way.

- The cost is enormous, reproducing this effort will be difficult despite the straightforward methodology; moreover GPT 3.5 / ChatGPT is ever evolving meaning the same dataset will not be reproducible even months from acceptance.

**Reproducibility:**

3: Could reproduce the results with some difficulty. The settings of parameters are underspecified or subjectively determined; the training/evaluation data are not widely available.

**Reviewer Confidence:**

4: Quite sure. I tried to check the important points carefully. It's unlikely, though conceivable, that I missed something that should affect my ratings.

---

> ### Author Rebuttal · Authors · 2023-08-28
>
> Thank you for appreciating our effort for developing and evaluating SODA, as well as the impressive performance of COSMO on the out-of-domain dataset!
>
> **1. Evaluations with Amazon Mechanical Turk and the evaluation criteria:**
> Good point! We acknowledge that evaluations by crowdworkers tend to be noisier than those by domain experts, who typically excel in many existing NLP tasks. However, determining the naturalness and coherency of social chitchats can be also done relatively easily by lay people as they engage in chitchats everyday. Furthermore, we run separate qualification tests to recruit high quality workers, and ask them to justify their choice in free-form text to prevent hasty judgments. For selecting the evaluation criteria, we follow existing works in the machine dialogue field [1, 2, 3].
>
> **2. You can now easily build SODA with much less cost.**
> The original cost for building SODA was under 40,000 USD, considerably less than the cost of crowdsourcing the same volume of conversation data. It is worth noting that the cost of using GPT-models has decreased significantly since we built the current version of SODA. The same amount of conversations can be generated for tenth the original cost. Moreover, our exciting preliminary tests with the newly released open-source Llama-2-13b-chat model show that we can obtain conversations comparable to those in SODA, by applying our CO_3 method.
>
> **3. Q1 - Do you think COSMO scores are elevated by the underlying conversations generated from GPT-3.5 in SODA?**
> Yes, however, the choice of LLM is not strictly limited to GPT-3.5. We attribute COSMO's high performance to its training on conversations grounded in a broad array of specific contexts. Many existing dialogue datasets do not cover wide ranges as they are more narrowly focused on specific themes (e.g., persona, empathy), and they lack scale as they are manually crafted by crowd-workers. Training models on conversations generated by our CO_3 method with other recent LLMs (e.g., Llama2) can also yield impressive performance.
>
> **4. Q2 - Can we expect GPT 3.5 to perform comparably to COSMO in naturalness and consistency in Table 5, thus having similar improvements over Blender, GODEL, Koala, Vicuna, and GT?**
> Yes, we expect GPT-3.5 to perform comparable to COSMO, and perform better than existing models [4].
>
>
> **5. Q3 - Do you think the selected metrics favor SODA's characteristic of detailed grounding backstories? Is SODA representative of human-bot or human-human conversations?**
> No, the selected metrics do not specifically favor conversations being grounded in backstories, as they are metrics widely used in dialogue evaluation (e.g., consistency, coherency, specificity, and naturalness) [1, 2, 3]. We would also like to note that conversations in DailyDialog and BlendedSkillTalk are also grounded in specific backstories and context, respectively.
>
> SODA represents human-human conversations. However, it also includes conversations involving assistance (e.g., clerk, customer service), which can potentially help improve human-bot interactions in AI assistant models
>
>
> **6. Q4 - How much did it cost to construct SODA?**
> Please see our response #2.
>
>
> **7. Q5 - What is the reason behind the discrepancy between humans and GPT 3.5 in validating the commonsense? How large is the holdout for this evaluation?**
> The discrepancy comes from using the full commonsense triples vs. the head event in the commonsense triples. During the human validation, workers were asked to check whether the full commonsense triple is included in the conversation. We used 100 random dialogue samples for human validation. On the other hand, for validation using GPT-3.5, we filtered based on the head event only.
>
>
> [1] Sashank Santhanam and Samira Shaikh. 2019. Towards Best Experiment Design for Evaluating Dialogue System Output. In Proceedings of the 12th International Conference on Natural Language Generation, pages 88–94, Tokyo, Japan. Association for Computational Linguistics.
>
> [2] Mehri, Shikib, et al. "Report from the NSF future directions workshop on automatic evaluation of dialog: Research directions and challenges." arXiv preprint arXiv:2203.10012 (2022).
>
> [3] Matthew Marge, João Miranda, Alan Black, and Alexander Rudnicky. 2010. Towards Improving the Naturalness of Social Conversations with Dialogue Systems. In Proceedings of the SIGDIAL 2010 Conference, pages 91–94, Tokyo, Japan. Association for Computational Linguistics.
>
> [4] Maximillian Chen, Xiao Yu, Weiyan Shi, Urvi Awasthi, and Zhou Yu. 2023. Controllable Mixed-Initiative Dialogue Generation through Prompting. In Proceedings of the 61st Annual Meeting of the Association for Computational Linguistics (Volume 2: Short Papers), pages 951–966, Toronto, Canada. Association for Computational Linguistics.

---

### Official Review · Reviewer_H9QN · 2023-08-11

**Soundness:** 4

**Excitement:**

4: Strong: This paper deepens the understanding of some phenomenon or lowers the barriers to an existing research direction.

**Paper Topic And Main Contributions:**

In this paper the authors introduce the SODA dataset, a large-scale social dialog dataset consisting of various situations based on a knowledge graph containing social commonsense knowledge. The main idea of the dataset is to use GPT-3.5 combined with the knowledge graph to generate narratives, which is again put into GPT-3.5 to generate conversations. The authors show that the quality of the dataset surpasses existing conversation datasets in several aspects. Next, the authors fine-tune a conversation model named COSMO based on this dataset, developed, which surpasses existing conversation-based LLMs.

**Questions For The Authors:**

- As according to Tables 4 and 12, SODA does seem to have more diversity in the types of emotions compared to different datasets. Yet, even in this dataset, the distribution is skewed towards particular emotions, where the top-4 emotions account for 40% of the entire dataset. What would your opinions be towards the idea of explicitly controlling for the types of emotions so that it could be more evenly distributed?
- While you did provide a per-unit cost of creating the dataset, it would be interesting if you could also disclose the total cost of creating the dataset from scratch, that is the total sum of the GPT models used at different stages.
- I wonder whether there is any potential bias in SODA due to the nature that the dataset is generated largely from GPT models. Specifically, whether the dataset is representative of different demographics, especially under-represented groups.

**Reasons To Accept:**

- The paper is well-motivated and well-presented.
- The deliverables of this paper - the SODA dataset and the COSMO model - can be valuable to the social NLP community. Especially, I think the validation effort put into both deliverables should be appreciated.
- This paper demonstrates a use case of LLMs and how they can be used to generate datasets that can be put to use. The combination of knowledge graphs with various prompting strategies shows that it is possible to generate a diverse set of social interactions that can be used to improve the social capabilities of existing LLMs.

**Reasons To Reject:**

- One concern is that this dataset is generated by GPT across multiple stages, and the biases that the model is known to have may be inherent in the dataset throughout various forms.Also, the close nature of GPT will make it difficult to reproduce the results.

**Reproducibility:**

4: Could mostly reproduce the results, but there may be some variation because of sample variance or minor variations in their interpretation of the protocol or method.

**Reviewer Confidence:**

4: Quite sure. I tried to check the important points carefully. It's unlikely, though conceivable, that I missed something that should affect my ratings.

---

> ### Author Rebuttal · Authors · 2023-08-28
>
> Thank you for recognizing the value of SODA and COSMO to the NLP community! We also appreciate your acknowledgment of our contribution in using “knowledge graphs to generate diverse social interactions”!
>
> **1. Our effort to minimize potential biases and include diverse groups in SODA:**
> Thank you for bringing up an important point! We also acknowledge that there can be biases related to the base LLM used for SODA. We prioritize filtering out harmful biases with a two-step process using a social norm based dialogue safety model and a safety-related API, removing about 4.5% of the conversations.
>
> We also fully agree with you on the importance of representing under-represented groups. To achieve this in SODA, we use the top 1,000 names in the US social security number (SSN) applicants list, which covers 95% names in the list from 1990 to 2021, including names commonly associated with diverse genders (including non-binary and gender-neutral names) as well as various ethnicities. Developing methods for thoughtfully integrating diverse cultural commonsense knowledge in our framework will also be an important future direction, as naively prompting LLMs to generate texts related to minorities can result in stereotypes and parodies of them [1].
>
>
> **2. You can reproduce SODA with other choice of LLMs too!**
> Although the current version of SODA is generated with a closed LLM, our CO_3 method is generally applicable to any LLMs. Our initial tests with the newly released Llama-2-13b-chat model, an open-source LLM, we are excited to find the conversations generated with our CO_3 framework also have impressive quality comparable to those in SODA. We plan to release our CO_3 code to inspire further research and to enable the distillation of diverse conversations across various LLMs.
>
>
> **3. Explicitly controlling for the types of emotions:**
> Thank you for a great suggestion! This is indeed possible using our method. Given that commonsense knowledge graphs also include emotional reaction knowledge, we can upsample triples with under-represented emotion and use our CO_3 method to generate more relevant conversations.
>
> Also, it is worth noting that the reason those four emotions are predominant across all three datasets is because many common responses, such as "How are you?", "That's great!", and "Thank you", are categorized as curiosity, admiration, and gratitude.
>
>
> **4. The cost of constructing SODA.**
> The original cost was less than 40,000 USD, which is significantly cheaper than the cost of crowdsourcing the same amount of conversation data. Moreover, the expense of using GPT-models has decreased significantly since we built SODA. We believe that now, the same volume of dialogue can be generated for tenth the original cost. We again kindly note that you can also apply our CO_3 method on open-source LLMs, such as LLaMA2, and obtain comparable conversations.
>
>
>
> [1] Myra Cheng, Esin Durmus, and Dan Jurafsky. 2023. Marked Personas: Using Natural Language Prompts to Measure Stereotypes in Language Models. In Proceedings of the 61st Annual Meeting of the Association for Computational Linguistics (Volume 1: Long Papers), pages 1504–1532, Toronto, Canada. Association for Computational Linguistics.

---

### Official Review · Reviewer_ZTzC · 2023-08-12

**Soundness:** 4

**Excitement:**

4: Strong: This paper deepens the understanding of some phenomenon or lowers the barriers to an existing research direction.

**Paper Topic And Main Contributions:**

This paper presents SODA, a new large scale synthetic social dialog dataset. In order to generate this dataset they start by verbalising triplets from a knowledge graph. These short sentences are then enriched by GPT-3.5 into a short narrative. Once having the enriched knowledge graph triplet, this is fed again into GPT-3.5 for creating a full conversation. After filtering the dataset they end up with 1.5M dialogues.

The dataset quality is evaluated by sampling 300 examples and comparing against human generated datasets taking 6 different into account. Overall they show that SODA is preferred by humans.

Using the SODA dataset they train the COSMO model that outperforms different SOTA models on the Daily Dialog dataset.

**Questions For The Authors:**

- Do you think that your dataset helps more due to scale or due to quality when comparing with SOTA datasets?

**Reasons To Accept:**

- They open sourced SODA a large dataset dialogue dataset.
- SODA shows to be preferred by humans when compared to previous dialogue datasets.
- COSMO, a model trained on SODA, outperforms SOTA models.

**Reasons To Reject:**

- SODA is synthetic and generated with a close model.
- Distilling GPT-3.5 models has shown to be efficient in many previous work.

**Reproducibility:**

4: Could mostly reproduce the results, but there may be some variation because of sample variance or minor variations in their interpretation of the protocol or method.

**Reviewer Confidence:**

3: Pretty sure, but there's a chance I missed something. Although I have a good feel for this area in general, I did not carefully check the paper's details, e.g., the math, experimental design, or novelty.

---

> ### Author Rebuttal · Authors · 2023-08-28
>
> Thank you for acknowledging the quality of SODA and COSMO, as well as our commitment to open-source!
>
> **1. SODA is generated with a closed model, but it can be reproduced with open-source LLMs too!**
> While the current version of SODA is created with a closed model, our CO_3 method for building SODA is applicable to other open-source LLMs as well, such as LLaMA. In our preliminary tests with the newly released Llama-2-13b-chat model, an open-source LLM, we are excited to find the conversations generated using our CO_3 framework also have great quality comparable to those in SODA. Furthermore, we would also like to note that our construction code for SODA will also be publicly available upon acceptance to encourage more diverse conversation distillations with various LLMs.
>
> **2. The primary contribution of our work is the step-by-step contextualization of commonsense knowledge graphs, rather than a naive straightforward distillation.**
> The primary contribution of our work is not merely in distilling conversations. Rather, it is in using the commonsense knowledge graph to obtain a much broader spectrum of social conversations with commonsense by contextualizing it step-by-step. In Section 3.3, we show that naively distilling conversations from the LLM leads to lack of quality compared to our approach. For example, conversations in SODA outperform conversations distilled without contextualization, scoring double in terms of specificity and interestingness.
>
> **3. Comparing SODA’s scale and quality with other datasets:**
> COSMO, which is trained on SODA, outperforms models trained on significantly larger conversation datasets (1.5M vs 1.5B) sourced from Reddit threads and web crawls. This indicates the contribution of SODA's quality to COSMO’s performance. Compared to crowd-sourced datasets, SODA contributes in both scale and quality. Therefore, SODA occupies a unique position by fulfilling both scale and quality criteria relative to existing datasets. An interesting potential future direction would be to explore experiments isolating the effects of scale and quality by training on smaller portions of SODA to see how much scale matters.

---

### Meta-Review · Area_Chair_ppN9 · 2023-09-16

**Recommendation:** 5

**Metareview:**

This paper presents SODA, a framework for generating social chit-chat dialogue data using pre-trained LLMs (GPT3.5 in this work). This framework is used to produce a 1.5M dialogue dataset and train a T5-based LLM (COSMO) which is evaluated. Both the code and model will be released as open-source. All reviewers appreciated the need for such a framework and the fact that the authors are willing to open-source both data and code. They also liked the fact that the authors used commonsense triplets for dialogue generation. Most concerns centered around cost and bias, given that GPT3.5 was used for data generation, and one reviewer pointed out lack of details and evaluations. All concerns, however, were properly addressed by the authors and I am happy to accept this paper.

---

### Decision · Program_Chairs · 2023-10-07

**Decision:**

Accept-Main

**Comment:**

This paper presents SODA, a framework for generating social chit-chat dialogue data using pre-trained LLMs (GPT3.5 in this work). This framework is used to produce a 1.5M dialogue dataset and train a T5-based LLM (COSMO) which is evaluated. Both the code and model will be released as open-source. All reviewers appreciated the need for such a framework and the fact that the authors are willing to open-source both data and code. They also liked the fact that the authors used commonsense triplets for dialogue generation. Most concerns centered around cost and bias, given that GPT3.5 was used for data generation, and one reviewer pointed out lack of details and evaluations. All concerns, however, were properly addressed by the authors and I am happy to accept this paper.